Characterisation of an atypical manifestation of black band disease on Porites lutea in the Western Indian Ocean

Séré Mathieu mathieu.sere@gmail.com 1 2 3
Wilkinson David A. 4
Schleyer Michael H. 2
Chabanet Pascale 3
Quod Jean-Pascal 1
Tortosa Pablo 4
1 Agence pour la Recherche et la VAlorisation Marines (ARVAM) , Saint-Denis , Réunion , France
2 Oceanographic Research Institute (ORI) , Durban , KwaZulu-Natal , South Africa
3 UMR ENTROPIE, Labex CORAIL, Research Institute for the Development (IRD) , Saint-Denis , Réunion , France
4 Unité Mixte de Recherche “Processus Infectieux en Milieu Insulaire Tropical” (UMR PIMIT), Université de La Réunion, Inserm1187, CNRS9192, IRD249, Plateforme de Recherche CYROI , Saint Denis , Réunion , France
Reimer James
Electronic publication date: 2016 Jul 6
Publication date: 2016
Volume: 4
Electronic Location ID: e2073
Received 2016 Jan 28; Accepted 2016 May 3
Copyright: ©2016 Séré et al.
Copyright year: 2016
Copyright holder: Séré et al.
License: This is an open access article distributed under the terms of the Creative Commons Attribution License, which permits unrestricted use, distribution, reproduction and adaptation in any medium and for any purpose provided that it is properly attributed. For attribution, the original author(s), title, publication source (PeerJ) and either DOI or URL of the article must be cited.
License URL: https://creativecommons.org/licenses/by/4.0/

Keywords: Bacterial communities, Coral disease, Metabarcoding, Porites black band disease, Reunion Island, Scleractinian corals

Funding: The European Union (EU, FEDER) The Regional Council of Reunion The French Ministry of Higher Education and Research (DRRT) The French Department of Ecology, Sustainable Development, Transportation and Housing (DEAL) The French Ministry of Overseas (MOM) The Western Indian Ocean Marine Science Association (WIOMSA) The South African Association for Marine Biological Research (SAAMBR) This work was co-funded by the European Union (EU, FEDER), the Regional Council of Reunion, the French Ministry of Higher Education and Research (DRRT), the French Department of Ecology, Sustainable Development, Transportation and Housing (DEAL), the French Ministry of Overseas (MOM), the Western Indian Ocean Marine Science Association (WIOMSA) and the South African Association for Marine Biological Research (SAAMBR). The funders had no role in study design, data collection and analysis, decision to publish, or preparation of the manuscript.

==============================
Recent surveys conducted on Reunion Island coral reefs revealed an atypical manifestation of black band disease on the main framework building coral, Porites lutea. This BBD manifestation (PorBBD) presented a thick lighter-colored band, which preceded the typical BBD lesion. Whilst BBD aetiology has been intensively described worldwide, it remains unclear if corals with apparently similar lesions across coral reefs are affected by the same pathogens. Therefore, a multidisciplinary approach involving field surveys, gross lesion monitoring, histopathology and 454-pyrosequencing was employed to provide the first comprehensive characterization of this particular manifestation. Surveys conducted within two geomorphological zones over two consecutive summers and winters showed spatial and seasonal patterns consistent with those found for typical BBD. Genetic analyses suggested an uncharacteristically high level of Vibrio spp. bacterial infection within PorBBD. However, microscopic analysis revealed high densities of cyanobacteria, penetrating the compromised tissue as well as the presence of basophilic bodies resembling bacterial aggregates in the living tissue, adjacent to the bacterial mat. Additionally, classical BBD-associated cyanobacterial strains, genetically related to Pseudoscillatoria coralii and Roseofilum reptotaenium were identified and isolated and the presence of sulfate-reducers or sulfide-oxidizers such as Desulfovibrio and Arcobacter, previously shown to be associated with anoxic microenvironment within typical BBD was also observed, confirming that PorBBD is a manifestation of classical BBD.

Introduction

Black band disease (BBD) is one of the most widespread (Richardson, 2004; Richardson et al., 2009), destructive (Gantar et al., 2011; Richardson et al., 2009; Sato, Bourne & Willis, 2009) and intensively studied diseases on coral reefs worldwide (Al-Moghrabi, 2001; Boyett, 2006; Boyett, Bourne & Willis, 2007; Dinsdale, 2002; Edmunds, 1991; Kuta & Richardson, 2002; Raymundo & Weil, 2015; Rodriguez & Croquer, 2008; Rützler, Santavy & Antonius, 1983; Sato, Bourne & Willis, 2009; Voss & Richardson, 2006; Zvuloni et al., 2009). Gross lesions of BBD are generally described (based on their presentation in the field) as a dark-coloured band (a few millimetres to centimetres wide, and up to 1 mm thick) separating living tissue from dead skeleton, and migrating across the coral colony (Antonius, 1981; Cooney et al., 2002; Gantar et al., 2011; Myers & Richardson, 2009; Raymundo & Weil, 2015; Rützler, Santavy & Antonius, 1983). As many as 70 coral species have been reported to be affected by BBD (Sutherland, Porter & Torres, 2004), particularly massive and slow-growing reef building corals (Gantar et al., 2011; Richardson, 2004). Factors affecting susceptibility of corals to BBD or enhancing its progression and spread in corals are still not fully understood (Aeby & Santavy, 2006; Boyett, Bourne & Willis, 2007; Rodriguez & Croquer, 2008; Sato, Bourne & Willis, 2009; Voss & Richardson, 2006; Zvuloni et al., 2009). However, a few experimental studies have linked nutrient enrichment, elevated temperature and light intensity to the pathogenesis of BBD in corals (Aeby & Santavy, 2006; Boyett, Bourne & Willis, 2007; Voss & Richardson, 2006).

Historically, BBD pathology was first microscopically described as a microbial consortium dominated by filamentous cyanobacteria associated with sulphate reducing (Garrett & Ducklow, 1975) and sulphide oxidizing bacteria (Rützler, Santavy & Antonius, 1983). Later, studies using culture-independent molecular techniques revealed a dense and diverse microbial community classified into four functional groups, comprising photoautotrophs (Cyanobacteria), sulphate reducers (Desulfovibrio), sulphide oxidizers (Beggiatoa) and organo-heterotrophs (Vibrio) (Cooney et al., 2002; Frias-Lopez et al., 2004; Myers, Sekar & Richardson, 2007; Richardson, 2004; Sekar et al., 2006; Viehman et al., 2006). Among these groups, a few bacteria have been suspected to be primary pathogens, including Desulfovibrio spp (Viehman et al., 2006) and Vibrio coralliilyticus (Arotsker et al., 2009); however, none of these species have been tested experimentally and/or satisfied Henle Koch’s postulates. In addition, variations have been detected in bacterial communities associated with BBD across geographic regions and between sympatric coral species (Voss et al., 2007). For instance, the presence of 16S rDNA sequences similar to Trichodesmium and Oscillatoria were reported in BBD-infected samples from Papua New-Guinea (Frias-Lopez et al., 2002), whereas members of the genera Geitlerinema, Leptolyngbya, Lyngbya, Oscillatoria, Phormidium, Pseudoscillatoria and Roseofilum were detected in BBD from the Caribbean, Philippines and Red sea (Casamatta et al., 2012; Myers, Sekar & Richardson, 2007; Rasoulouniriana et al., 2009; Sekar et al., 2006). More recently, an early stage of BBD has been identified, named cyanobacterial patch (CP), where bacterial communities are initially rich in Bennothrix sp. before being progressively displaced by bacteria related to Oscillatoria sp. (Sato, Bourne & Willis, 2009; Sato, Willis & Bourne, 2010). The high variability in BBD bacterial communities found between localities and infected host species may indicate that BBD actually derives from an earlier infection, which favours the infection and subsequent proliferation of opportunistic microorganisms such as cyanobacteria. However, beyond this highly speculative assumption and despite being intensively studied worldwide, the mechanisms of BBD development remain unclear and no primary pathogens have yet been clearly identified.

Recent surveys, conducted on western Indian Ocean (WIO) coral reefs (Fig. 1) over two consecutive summers and winters between 2010 and 2012 (Séré et al., 2015), revealed an atypical manifestation of black band disease on two of the main framework building corals, Porites lobata and Porites lutea, and hereafter referred to as “Porites black band disease” (PorBBD). Following standardized terminology (Work & Aeby, 2006), PorBBD is characterized by a diffuse, central or peripheral, undulating to smooth, gray to black band, leaving behind dead skeleton (Fig. 2). The older exposed skeleton is progressively colonized by endophytic algae. In contrast with typical BBD, PorBBD exhibits a lighter black band and a thin to medium (0.5–2 cm in width), undulating to smooth white band of bleached tissue separating the healthy tissue from the black band itself. Our study aimed at providing a comprehensive characterization of PorBBD using a multidisciplinary approach involving field surveys, gross lesion monitoring, and description of histopathologic features together with a description of the associated bacterial diversity.

Figure 1 Study sites in Reunion Island.

Map showing the study sites in Reunion Island, Western Indian Ocean.

Figure 2 Massive colonies of Porites lutea exhibiting signs of Porites black band disease (PorBBD) at (A) Ravine des Poux, (B) and (C) La Corne, and (D) Trou d’Eau in Reunion Island.

Ds, dead skeleton; Bcy, black cyanobacteria; Wf, white front; Ht, healthy tissue.

Material and Methods

The sampling of Porites lutea colonies for this study was authorised by the French Department of Ecology, Sustainable Development, Transportation and Housing (DEAL), and CITES (Permit no. FR1197400391-FR1197400394-1)

Field surveys and progression rate

Surveys were undertaken in Reunion at four latitudinal sites on the outer reef slope and reef flat following protocols adapted to these geomorphological zones. The outer reef slope is characterized by a succession of spurs and grooves that represent different habitats. Spurs are covered mainly with hard corals, whereas grooves are often filled with sand and coral rubble. In order to stay within the coral community, five 10 m × 2 m belt-transects were laid along the different spurs at the same depth. Surveys on the inner reef flat were conducted along three 20 m × 2 m belt-transects positioned parallel to the coastline in order to avoid crossing different coral communities. Transects were randomly laid and all starting points were geo-referenced. Details of the sites are given in Table 1. Surveys were conducted over two consecutive summers (December 2010–January 2012) and winters (September 2010–October 2011) to gain a measure of seasonality in the prevalence of PorBBD. Averaged sea surface temperatures ranged from 23.5°C in winter to 31.6°C in summer in Reunion. All massive P. lutea and P. lobata displaying signs of PorBBD were counted along each transect. P. lutea is a common coral on Reunion Island coral reefs and is mostly found on the reef flat (0.5–1.5 m deep). Colonies are generally brown, yellow-brown or yellow green in colour with corallites filled with skeletal elements. In contrast, P. lobata is relatively rare on the reef flat and commonly forms helmet-shaped colonies with lobed upper surfaces. Colonies are mainly purple-blue and the corallites have relatively few skeletal elements (G Faure, pers. comm., 2015). The prevalence of PorBBD was estimated as = ((the number of PorBBD-infected colonies)/(the total number of massive Porites > 2 cm) × 100), counted along each transect in 1 m × 1 m quadrats (20–40 quadrats per transect). Finally, disease fronts were monitored on a monthly basis from 30 November 2010 to 10 December 2011 in order to follow PorBBD progression. Nails were driven into the dead portions of five Porites lutea colonies behind disease front as reference markers for this purpose. The progression rate was recorded as the linear distance between the nails and the nearest live tissue using photographs.

Table 1 Location and depth of the reef sites and stations selected for this study.

Sites	Stations	Habitat	Reef depth (m)	Latitude/longitude	
L’Ermitage	3-Chameaux	Reef flat	0.5–1.0	–21.080351°; 55.219576°	
La Saline	Trou d’Eau	Reef flat	0.5–1.0	–21.103312°; 55.242294°	
Saint-Leu	La Corne	Reef flat	0.5–1.0	–21.165960°; 55.285080°	
Saint-Leu	Ravine des Poux	Reef flat	0.5–1.0	–21.176397°; 55.285985°	
L’Ermitage	3-Chameaux	Reef slope	10.0–12.0	–21.081281°; 55.217590°	
La Saline	Trou d’Eau	Reef slope	10.0–12.0	–21.106160°; 55.239540°	
Saint-Leu	La Corne	Reef slope	10.0–12.0	–21.165940°; 55.281930°	
Saint-Leu	Ravine des Poux	Reef slope	10.0–12.0	–21.175490°; 55.283460°	

Histopathology

Samples of Porites lutea exhibiting signs of PorBBD were collected using SCUBA and snorkelling. Core samples (10 mm core tubes) were taken from three healthy (HT) and five diseased (DT) tissues, and fixed in 4% formalin for histological examination of their tissue structure. Diseased tissue (DT) was sampled at the lesion boundary interface separating dead tissue from healthy tissue (HT), while control tissue (CT) was cored from an apparently uninfected colony. All DT, HT and CT samples were then coated in 1.5% (w/v) agarose to retain the spatial integrity of the tissues. They were then decalcified using 1% HCl and EDTA renewed every 12 h until process completion. Decalcified tissues were finally dehydrated in a gradient of ethanol baths, cleared with xylene and embedded in paraffin wax. Cross sections of 6–8 µm thick were cut using a microtome, mounted on glass slides and stained with Harris haematoxylin and eosin containing phloxine B as previously described for the diagnosis of tissue fragmentation, necrosis and the identification of invasive organisms (Sudek et al., 2012; Work & Aeby, 2011). Serial sections were examined under a light microscope and photographed using NIS Element software (Nikon©).

Cyanobacterial culturing, isolation, and identification

Fresh PorBBD mat was collected from four infected Porites lutea colonies using needles and sterile syringes to identify the dominant cyanobacterial strains. Samples were placed in 10 ml centrifuge tubes with seawater and held in darkness at 20°C until the return to the laboratory. Cyanobacterial filaments (one when possible) were isolated from the raw samples under a light microscope and transferred to agar plate containing Z8 medium (Kotai, 1972) enriched with NaHCO3, (NH4)2SO4 and Vitamin B12. Inoculated plates were incubated at 27°C with a 12 h light:dark photoperiod and constant irradiance of 20 µmol photons m−2 s−1. Bacterial strains were routinely passaged between petri dishes in order to obtain clonal isolates for use in molecular analysis. Genomic DNA was then extracted from each cyanobacterial isolate by boil lysing in 100 µL of 5 mM Tris/HCl at 100°C for 5 min. PCR amplification was carried out in a volume of 25 µl GoTaq®Hot Start Green Master Mix (Promega, Madison, WI) containing 0.5 mM of 16S rRNA gene cyanobacterial primers CYA781R/CYA106F (Nübel, Garcia-Pichel & Muyzer, 1997; Rasoulouniriana et al., 2009; Sussman, Bourne & Willis, 2006) and 10 ng of template DNA. Amplification conditions for the PCR included an initial denaturing step of 5 min at 94°C, followed by 35 cycles at 94°C for 60 s, 60°C for 60 s, 72°C for 60 s, and a final extension step of 5 min at 72°C. Sequences obtained for each cyanobacterial strain were examined for error and edited using GENEIOUS™ Pro (V.5.6.6) sequencing software (Kearse et al., 2012). All consensus sequences were submitted to BLAST at the National Centre for Biotechnology Information (NCBI, www.ncbi.nlm.nih.gov) and compared with published sequences. The 16S rRNA sequences were aligned with reference sequences from closely related to known cyanobacterial strains available in genbank. A phylogenetic tree was built by neighbour-joining in GENEIOUS™ Pro (V.5.6.6) with bootstrap values based on 1,000 replicates.

Metagenomic profile of bacterial 16S rRNA genes

Sample collection and DNA extraction

Samples of Porites lutea were collected from healthy (HT) and diseased (DT) sections of two infected P. lutea colonies as well as a single control sample (CT). DT were sampled from the lesion boundary interface with visually healthy tissue (HT), and the sample of CT was taken from completely asymptomatic coral colonies. Cores of DT, HT and CT (2.2 cm diameter to a depth of 0.5–1 cm) were collected using a sterile stainless steel core tube and placed individually in sterile disposable 50 ml polypropylene centrifuge tubes and kept under low light conditions at 2°C in a cool box. Immediately upon return to the laboratory, the seawater within each tube was decanted and the coral samples were immersed in absolute ethanol and stored at –80°C for molecular analysis. To ensure that we had collected P. lutea, all samples were examined under the microscope prior their preservation in absolute ethanol.

Bacterial genomic DNA was extracted from CT, HT and DT using the NucleoSpin® Soil Kit (NucleoSpin Extract II, Macherey-Nagel, Düren, Germany). Approximately 150 mg of both tissue and skeleton were scratched from the core surface using a sterile scalpel blade, placed in a 1.5 ml centrifuge tube with 700 µl of lysis buffer and crushed using a fresh disposable plastic rod. Samples were then placed in lysing matrix tubes for DNA extraction. The DNA was eluted with 50 µl sterile elution buffer and its quality was verified by electrophoresis in agarose gels (1.5% wt/vol) stained with GelRed™ (Biotium Inc., Hayward, California, USA).

PCR and 454 pyrosequencing

The composition of bacterial communities associated with CT, HT and DT samples was analysed using 454-pyrosequencing technology (Roche, Nutley, NJ, USA) at GENOSCREEN (Campus de l’Institut Pasteur de Lille, France). In our design, 16S rRNA variant regions V3 and V4 were amplified using forward (TACGGRAGGCAGCAG) and reverse (GGACTACCAGGGTATCTAAT) primers. These primers were linked to 5′ with MID tags, a GsFLX key and GsFLX adapters. Each sample was amplified independently twice with distinct MID tags, allowing the identification of each gene pool. Quality control was performed using the Agilent DNA 100 (Agilent Technologies). The quantity of each PCR product was measured with Picogreen and all products were mixed in equimolar concentrations prior to 454 GsFLX sequencing.

Sequence analyses

All 454 GSFLX sequences were sorted by MID identification using GENEIOUS™ Pro (V.5.6.3). All generated reads were analysed using the SILVA online NGS tool (available online at www.arb-silva.de/ngs). Raw sequence reads were aligned with a gap extension penalty of 2 and a gap penalty of 5. Reads were filtered based on the following quality criteria: minimum length—200 bp, minimum quality score—30, maximum percent ambiguities—1%, minimum base pair score—30 and maximum percent repetitive—2%. Remaining reads were clustered into operational taxonomic units (OTUs) at a threshold sequence identity of 99%. OTUs were classified by BLAST score comparison against the SILVA rRNA database version 115, with a classification similarity threshold of 93%. Data from the SILVA classification were exported for further processing in MEGAN software version 5.0.78 beta (Huson et al., 2007) using the lowest common ancestor (LCA) algorithm with all parameters kept at default values (min support, 5; min score, 35; top percent, 10.0; win score, 0.0). Cyanobacterial 16S rRNA gene sequences are accessible through the NCBI GeneBank database under accession numbers KF957835–KF957838. Raw 454-pyrosequencing reads were submitted in Study (BioProject PRJNA231011) to the NCBI Sequence Read Archive (SRA).

Statistical analysis

Prevalence of PorBBD was calculated per transect and for each site. Data were tested prior to analysis for homoscedasticity (Levene’s test) and normality of variance (Kolmogorov–Smirnov and Lilliefors tests) and were then log transformed [log10(X)] for analysis of variance (ANOVA). Variations in the prevalence of PorBBD over the two survey years in the consecutive summers and winters and across reef zones (reef slope vs. reef flat) were tested using Factorial ANOVA (STATISTICA 8). Fisher tests were performed for post hoc multiple comparison. Rarefaction curves were performed on bacterial populations associated with each tissue category. Principal coordinate analysis (PCoA) and cluster analysis were performed using the Bray–Curtis similarity coefficient to compare the bacterial community structures of the different tissue categories and the Simpson’s and Shannon’s diversity indices were calculated. Finally comparisons of average values of bacterial communities associated with different tissue categories were performed using the t-test in STATISTICA. Before analysis, all OTU counts were normalized to avoid bias due to differences in the number of sequences obtained from each sample (Mitra, Klar & Huson, 2009).

Results

PorBBD prevalence and virulence

A total of 3,520 m2 of reef and 5,363 massive colonies of Porites lutea and Porites lobata were surveyed between September 2010 and January 2012. PorBBD varied seasonally and between the two geomorphologic reef zones. For instance, colonies of Porites exhibited significantly higher PorBBD prevalence on the reef flat (ANOVA: F = 1.18, p < 0.01), affecting an average of 4.1 ± 2.0% (mean ± SE) colonies compared to those observed at the deeper sites (0.2 ± 0.4; mean ± SE). The percentage of infected colonies recorded on both the reef flat and reef slope was significantly higher during summers than winters (ANOVAwinter1vs.summer1: F = 3.89, p < 0.01; ANOVAwinter2vs.summer2: F = 0.6, p < 0.05). The rate of tissue mortality measured on five massive colonies of P. lutea between 2011 and 2012 was 4.4 ± 0.12 (mean ± SE) mm day−1. Among the monitored colonies, two died approximately 13 months after the beginning of the study.

Microscopic characterisation of PorBBD

Comparison of cross sections of PorBBD-infected P. lutea colonies (Fig. 3A) revealed the presence of three distinct tissue regions; the first one being the oldest area of infection comprising dead and degraded tissue associated with cell debris, endophytic algae and other organisms such as cyanobacteria and ciliates (Fig. 3C). The second region was characterised by a mat of microorganisms, where filamentous cyanobacteria were clearly visible (Fig. 3C) perforating the compromised and dead tissues (Figs. 3E and 3F). Finally, the discoloured portion of the tissue next to the black band contained granular pigmented cells in both the epidermis and gastrodermis (Fig. 3B). Basophilic bodies resembling bacterial aggregates were also observed in this tissue region and were regularly surrounded by the same granular cells.

Figure 3 Histological sections of Porites lutea: (A) Ravine des Poux, (B) and (C) La Corne, and (D) Trou d’Eau in Reunion Island.

Ds, dead skeleton; Bcy, black cyanobacteria; Wf, white front; Ht, healthy tissue.

Identification of the dominant cyanobacterial strain

Cyanobacterial strains isolated from four individual PorBBD-infected Porites lutea formed dense clumps of brown filaments that were able to colonize an entire petri dish surface (75 cm2 of Z8 solid medium) in a single week (Fig. 4A). The four strains were motile and appeared morphologically similar with pointed, arrow-like calyptra (Fig. 4B). The isolates were genetically similar to each other (>98% identity) and phylogenetically affiliated to the cyanobacteria Pseudoscillatoria coralii (FJ210722) and Roseofilum reptotaenium (HM048872) (Fig. 4C).

Figure 4 Cyanobacteria retrieved from Porites black band disease (PorBBD).

(A) Clumps of brown cyanobacterial filamments (cy) growing in a petri dish with Z8 medium. (B) Photomicrograph of the cyanobacterial strain CYPBD1, closely related to the cyanobacterium Pseudoscillatoria coralii (FJ210722) and Roseofilum reptotaenium (HM048872), isolated from pure cultures. Note the pointed terminal cells called calyptra (cal). (C) Neighbour-joining phylogenetic tree showing the relatedness of the strains CPPORBBD1, CPPORBBD2, CPPORBBD3, and CPPORBBD4 with reference cyanobacterial strains. Numbers at each node are bootstraps values (%) obtained after 1,000 iterations.

Comparison of bacterial community structure of DT, HT and CT

Following Roche 454-pyrosequencing, a total of 52,257, 38,778 and 21,309 sequence reads were obtained from PorBBD infected tissue (DT1a-DT1b and DT2a-DT2b), apparently healthy tissue (HT1a-HT1b and HT2a-HT2b) and healthy (control) tissue (CTa and CTb) respectively (Table 2). The number of OTUs obtained for each sample categoy are summerised in Table 2.

Table 2 Total sequences read before and after sequence trimming, number of bacterial classes and genera and diversity indices for each sample and subsample of PorBBD (DT1a-DT1b and DT2a-DT2b), apparently healthy tissue (HT1a-HT1b and HT2a-HT2b) and control tissue (CTa and CTb).

	DT1a	DT1b	DT2a	DT2b	HT1a	HT1b	HT2a	HT2b	CTa	CTb	
Σ raw sequences	11,320	12,703	14,848	13,386	15,660	14,646	19,008	15,221	12,563	8,746	
Σ OTU	2,560	2,569	2,338	2,363	1,417	1,334	1,484	1,149	1,160	801	
Σ bacterial Phylum	14	8	12	10	6	8	13	12	14	11	
Σ genera	118	48	63	69	15	19	29	22	29	32	
Shannon index	4.689	4.586	4.477	4.509	3.333	3.398	3.204	3.191	3.563	3.666	
Simpson reciprocal index	13.642	13.334	13.458	13.650	9.090	9.294	8.336	8.326	9.949	10.215	

Bacterial community structures from different tissue samples were compared with Cluster analysis in MEGAN (V5.0.77) based on Bray-Curtis similarity matrices (Fig. 5A). Results revealed three clearly separated groups; Cluster 1 grouping all DTs, Clusters 2 and 3 grouping HT and CT samples (Fig. 5A). Rarefaction curves (Fig. 5B) nearly paralleled the x axis for the majority of samples, indicating that the overall bacterial diversity had likely been exhaustively characterized within the bounds of amplicon bias. Bacterial diversity estimated with Simpson’s and Shannon’s diversity indices showed that DT had higher bacterial diversity than both HT and CT (Table 2).

Figure 5 Bacterial community structures based on the classification of partial 16S RNA genes obtained from Porites black band disease lesions and healthy colonies of Porites lutea using MEGAN.

(A) Relative abundance (%) of bacterial classes and (B) number of bacterial genera assigned to the diverse classes associated with PorBBD-infected tissues (DT1 and DT2), healthy tissues (HT1 and HT2) and control tissue (CT). a and b distinguish the duplicate samples.

Diversity of bacterial community associated with PorBBD

Bacterial communities associated with PorBBD tissues were comprised of 8–14 different bacterial classes, dominated by γ-proteobacteria, α-proteobacteria, Bacteroidetes, δ-proteobacteria, ε-proteobacteria, Firmicutes and Cyanobacteria (Fig. 6A). In DT1, the α-proteobacteria (34.5%) and γ-proteobacteria (32.36%) were most commonly represented, followed by Bacteroidetes (11.26%), δ-proteobacteria (10.62%), Firmicutes (4.1%), ε-proteobacteria (3.94%), and Cyanobacteria (0.3%) classes. DT2 was dominated by γ-proteobacteria (60.0%), ε-proteobacteria (11.45%), Bacteroidetes (9.04%), Firmicutes (6.7%), δ-proteobacteria (5.2%), α-proteobacteria (3.0%) and Cyanobacteria (0.4%) classes. At a higher taxonomic level of MEGAN’s cladogram, 48–118 genera were obtained from DT (Table 2) with the α-proteobacteria and γ-proteobacteria classes exhibiting the highest diversity (Fig. 6B). The most dominant genera observed in DT were Vibrio (13.5–33.9%), Desulfovibrio (4.8–13.2%), Alteromonas (3.2–15.5%), Arcobacter (4.8–12.1%), Glaciecola (4.1–10.2%), Salinimonas (2.1–5.0%), Ruegeria (0.3–5.5%), Algicola (0.3–4.4%) and Oscillatoria (0.3–0.6%).

Figure 6 Comparative analysis of bacterial communities associated with three tissue categories of Porites lutea.

(A) Cluster diagram and (B) rarefaction curves of bacterial communities associated with HT 1–2, healthy tissue, CT, control tissue and DT 1–2, diseased tissue of Porites lutea, created using MEGAN software version 5.0.78 beta. Numerals a and b distinguish the duplicate samples.

Diversity of bacterial community associated with HT and CT

Bacterial communities of HT and CT samples were almost exclusively dominated by the γ-proteobacteria class, comprising 98.2%, 96.9% and 95.9% of the total OTUs, respectively (Fig. 6A). Additional OTUs attributed to α-proteobacteria, δ-proteobacteria, Actinobacteria, β-proteobacteria and ε-proteobacteria were also found in common but at very low percentages. A total of 19–29 genera were obtained from the HT samples including Endozoicomonas (76.7–96.5%), Vibrio (0.27–18.7%), Photobacterium (0.41–1.07%), Acinetobacter (0.14–0.38%) and Pseudomonas (0.08–0.43%). Finally, OTUs obtained from CT were mainly represented by Endozoicomonas (59.0–63.4%), Vibrio (30.8–31.1%), Photobacterium (1.8–2.6%) Acinetobacter (0.5%) and Propionigenium (0.2–0.3%).

Figure 7 Venn diagram of bacterial genera showing their distributions in PorBBD-infected tissues (DT), healthy tissues (HT) and control tissue (CT).

∗ indicates potentially pathogenic bacterial genera as identified in a literature survey of coral diseases.

Comparative analysis of bacterial communities associated with DT, HT and CT

The average number of OTUs affiliated to bacterial genera in DT was significantly different from those in HT (t-test: df = 2 p < 0.0001) and CT (t-test: df = 2 p = 0.0001). No significant difference was obtained between HT and CT (t-test: df = 2 p = 0.99). In total, 29 different bacterial genera were found only in DT (Fig. 7), 28 genera were uniquely found in HT, and 19 genera could be found in both DT and HT. Among OTUs obtained only from DT, the most represented bacterial genera were Glaciecola (4.1–10.2%), Salinimonas (2.1–5.0%), Amphritea (1.0–1.6%), Rhodobacter (0.2–2.3%), Shimia (0.4–1.5%) and Oscillatoria (0.3–0.6%). Of note, the genus Ruegeria, Desulfovibrio, Arcobacter, Alteromonas, Aestuaribacter, Salinimonas and Algicola were highly represented in DT but were also found in common with HT and/or CT at very low percentage. The full distribution of bacterial genera per tissue sample is depicted in Fig. 7 and Table 2.

Discussion

This study constitutes the first characterisation of an atypical form of BBD found on Reunion coral reefs, Porites black band disease (PorBBD). Surveys conducted on two geomorphological reef zones revealed spatial variability, with more infected colonies on the reef flat (0.5–1 m) than the reef slope (10–20 m). Similar patterns were previously reported in Florida Keys (Kuta & Richardson, 2002), the Republic of Maldives (Montano et al., 2012) and southern India (Thinesh, Mathews & Edward, 2009; Thinesh, Mathews & Patterson Edward, 2011) where typical BBD is more abundant at shallow than deep sites. This may be due to the proximity of the Reunion reef to the coastline (±500 m wide), where it is constantly exposed to high and increasing anthropogenic stress from sewage discharges, land-based pollution and eutrophication with compounds such as nitrates, ammonium, and phosphate (Chazottes et al., 2002; Naim, 1993). Several studies suggest that nutrient enrichment, sewage discharge and runoff may facilitate and increase disease outbreaks by enhancing the pathogen virulence and/or impairing host resistance (Haapkylä et al., 2011; Rodriguez & Croquer, 2008; Voss & Richardson, 2006).

Seasonal variations were also observed between 2010 and 2012: the average prevalence of PorBBD recorded on the reef flat peaked in summer 2010 (8.7 ± 2.8%; mean ± SE), dropped in winter 2011 (1.4 ± 1.0%; mean ± SE) and then increased again the following summer (7.7 ± 2.5%; mean ± SE). These results are consistent with those previously found during BBD surveys on the Great Barrier Reef in Australia (Boyett, Bourne & Willis, 2007; Sato, Bourne & Willis, 2009) and in Venezuela (Rodriguez & Croquer, 2008), the Red Sea (Zvuloni et al., 2009) and on Caribbean reefs (Edmunds, 1991). Seasonal fluctuations in BBD prevalence have been generally assumed to be driven by high light intensities and summer sea temperatures, which may reduce host resistance or/and increase pathogen virulence (Boyett, Bourne & Willis, 2007; Edmunds, 1991; Kuta & Richardson, 2002; Richardson & Kuta, 2003; Rodriguez & Croquer, 2008; Rützler, Santavy & Antonius, 1983; Sato, Bourne & Willis, 2009; Sato, Willis & Bourne, 2010). Richardson & Kuta (2003) showed that the association of high light and elevated temperatures promote the growth and progression of the cyanobacterium Phormidium corallyticum forming a dense mat. This BBD bacterial mat favouring the growth of sulphate reducers (e.g., Desulfovibrio species) and sulphide oxidizers (e.g., Beggiatoa species) generates anoxic conditions harmful to adjacent coral tissues. However, the hot season (December to March) in Reunion is also associated with heavy rainfall, high levels of ground water infiltration and surface water runoff, leading to an increased level of pollutants such as pesticide, fertiliser, sewage from septic systems and waste water in lagoon waters (Chazottes et al., 2002; Naim, 1993).

Field monitoring performed on tagged colonies confirmed the virulence of PorBBD with a mean tissue mortality rate reaching 4.4 ± 0.12 mm day−1 (mean ± SE). This progression rate is similar to values previously reported on typical BBD infected scleractininan corals in the Florida Keys (Kuta & Richardson, 1997), Australia (Boyett, Bourne & Willis, 2007; Sato, Willis & Bourne, 2010), Indonesia (Haapkylä et al., 2009) and India (Borger & Steiner, 2005; Thinesh, Mathews & Edward, 2009). Importantly, PorBBD demonstrated a high destructive potential for Reunion reefs with full mortality observed for 2 out of 5 colonies monitored over a one-year period.

Cross sections performed on PorBBD-infected tissues showed cyanobacterial aggregates similar to those previously observed in several studies of typical BBDs (Ainsworth et al., 2007; Barneah et al., 2007; Bythell et al., 2002; Sato, Bourne & Willis, 2009). The 16S sequences of cyanobacteria associated with PorBBD were closely related to cyanobacteria previously identified as Pseudoscillatoria coralii (Rasoulouniriana et al., 2009) and Roseofilum reptotaenium (Casamatta et al., 2012) isolated on different coral hosts from the northern Red Sea and the Caribbean, as well as with cyanobacteria isolated from BBD colonies in the central Great Barrier Reef (Sato, Willis & Bourne, 2010). Interestingly, this isolate, in contrast to cyanobacterial strains described in previous studies had unusually pointed terminal cells, or “calyptra”, suggesting the presence of a different cyanobacterial species associated with PorBBD.

Microscopic analysis also revealed the presence of filamentous cyanobacteria in both dead and compromised tissues. The ability of cyanobacteria to penetrate coral tissue has been demonstrated to play an important role in typical BBD investigations (Ainsworth et al., 2007; Barneah et al., 2007; Sato, Willis & Bourne, 2010). For instance, recent studies have suggested that the calyptra may be involved in tissue invasion (Ainsworth et al., 2007; Kramarsky-Winter et al., 2014), possibly via the secretion of toxins or other compounds (Miller & Richardson, 2012; Mydlarz, McGinty & Harvell, 2010; Whitton, 2008). The precise mechanisms of coral invasion by cyanobacteria are however unknown and require further investigation.

Granular and pigmented cells were found in high densities in PorBBD-infected tissue of P. lutea. They were found mainly in DT and have been proposed to result from an immune response (Mydlarz, McGinty & Harvell, 2010; Palmer, Mydlarz & Willis, 2008). Additionally, basophilic bodies resembling bacterial aggregates similar to those observed in Porites white patch syndrome (Séré et al., 2013) were found in the discoloured tissue adjacent to the bacterial mat but were not observed in HT. Thus, it is possible that other bacteria may promote PorBBD by initiating a primary infection that impairs the immune processes in corals and promoting progression of cyanobacteria (Miller et al., 2011). However, no evidence of direct physical destruction resulting from these basophilic bodies could be detected in our histological sections.

Bacterial community analysis via V3–V4, 16S metabarcoding suggested that the number of bacterial taxa identified in this study was higher than in other metagenomic analyses of bacterial communities associated with scleractinian corals (Littman, Willis & Bourne, 2011; Wegley et al., 2007). While HT and CT mostly contained γ-proteobacteria, DT yielded bacterial sequences from γ-proteobacteria (47.9 ± 2.44%; mean ± SE), α-proteobacteria (19.3 ± 4.2%; mean ± SE), Bacteroidetes (10.5 ± 0.4%; mean ± SE), δ-proteobacteria (8.2 ± 1.1%; mean ± SE), ε-proteobacteria (8.0 ± 1.6%; mean ± SE), Firmicutes (5.6 ± 0.7%; mean ± SE) and Cyanobacteria (0.4 ± 0.2%; mean ± SE). This result is not similar to patterns generally observed in other studies that have characterised microbial communities from typical BBDs (Arotsker et al., 2009; Cooney et al., 2002; Frias-Lopez et al., 2002; Sato, Willis & Bourne, 2013; Sekar et al., 2006). For instance, α-proteobacteria has been reported as the most represented and diverse class associated with BBD affecting scleractinian corals in several distant locations (Arotsker et al., 2009; Barneah et al., 2007; Miller & Richardson, 2011; Sekar et al., 2006), whereas γ-proteobacteria was the dominant class in PorBDD. In addition, Cyanobacteria sequences represented 4–25% of characterized bacteria from typical BBD lesions (Arotsker et al., 2009; Barneah et al., 2007; Sato, Willis & Bourne, 2013; Sekar et al., 2006) while only 0.4–0.6% of bacterial sequences from PorBBD had cyanobacteria origins. This is consistent with the PorBBD phenotype, which has a lighter band colour, compared to typical thick dark BBD. However, our further use of analytic tools such as ProbeMatch (Kim et al., 2009) suggests that these results should be interpreted with caution due to the inherent bias in taxon identification by PCR amplification due to the affinity of the primers targeting the V3–V4 region.

Among the major classes found in this study, the γ-proteobacteria Vibrio was the dominant genus in PorBBD-infected tissues representing 23.1 ± 2.1% (mean ± SE) of the overall OTUs. Several members of this genus have been identified as pathogens of corals, their virulence being attributed to enzyme secretions that initiate tissue penetration and degradation (Ben-Haim & Rosenberg, 2002; Ben-Haim et al., 2003; Rosenberg & Falkovitz, 2004). This study revealed the presence of OTUs attributed to Vibrio that are known to be highly proteolytic. However, since vibrionic OTUs were also abundant in non-infected tissues, their role in PorBBD in P. lutea needs to be individually assessed using multidisciplinary approaches combining bacterial culturing and inoculation/infection trials (Henle-Koch’s postulates).

The next most represented bacteria in PorBBD samples were Desulfovibrio and Arcobacter, accounting for 8.6 ± 1.5% (mean ± SE) and 7.9 ± 1.3% (mean ± SE) of the overall OTUs, respectively. These Proteobacteria have previously been found in BBD-infected corals from different locations (Frias-Lopez et al., 2002; Sato, Willis & Bourne, 2013) and seem to be important contributors to BBD aetiology, producing sulfated compounds suspected to exacerbate microbial virulence (Sato, Bourne & Willis, 2009). Although a very low percentage of these sulfide-reducing bacteria was found in healthy tissues, their relatively high abundance in PorBBD tissues suggests a tropism towards anoxic micro-environments (Cooney et al., 2002; Glas et al., 2012; Sato, Willis & Bourne, 2010; Sekar, Kaczmarsky & Richardson, 2008; Sekar et al., 2006; Viehman et al., 2006) rich in carbon compounds derived from cell debris and other organic nutrients produced during coral tissue lysis (Viehman et al., 2006). Other predominant genera affiliated to Glaciecola, Salinimonas, Amphritea, and Shimia were recorded only in PorBBD-infected corals but have not been reported in typical BBDs. Interestingly, Shimia was recently associated with a newly-reported Porites white patch syndrome on western Indian Ocean reefs (Séré et al., 2013), however no evidence of its pathogenicity has been established in this study. Importantly, no OTUs affiliated to potential pathogenic bacteria including Cytophaga, Clostridium, and Campylobacter, which generally occur in typical BBDs (Cooney et al., 2002; Frias-Lopez et al., 2004; Frias-Lopez et al., 2002; Sato, Willis & Bourne, 2013; Sato, Willis & Bourne, 2010) lesions were retrieved in PorBBD samples. Finally, no sulfide-oxydizing Beggiatoa related OTUs were retrieved from diseased samples while they have been commonly reported in typical BBDs (Sekar, Kaczmarsky & Richardson, 2008), but this again may be explained by the bias introduced by amplifying primer sequences, primer mismatches or limiting DNA template concentrations that have previously been reported to impair PCR (Frias-Lopez et al., 2002; Sekar et al., 2006).

In summary, the importance of PorBBD for Indian Ocean coral reefs should not be underestimated due to its potential for rapid progression in slowly growing reef-building corals and the associated high mortality rate. This disease shows several similarities with classical BBD manifestations, for example, the spatial and temporal patterns and propagation of PorBBD seem to be in agreement with those influencing typical BBDs. Moreover, cyanobacterial filaments in PorBBD are genetically closely related to those identified elsewhere. However, PorBBD exhibits key features that differ from typical BBD manifestations: firstly, an apparent low level of cyanobacterial infection produces a sparse black band, which is preceded by a discoloured tissue region. Basophilic bodies resembling bacterial aggregates were found in the discoloured tissue region, suggestive of an independent primary infection. Finally, an atypical composition of the associated bacterial communities suggests differing aetiology of PorBBD. However the presence of sulfate-reducers or sulfide-oxidizers, previously shown to be associated with an anoxic microenvironment within typical BBD, confirms that PorBBD is a manifestation of BBD. Further investigations, towards infection trials, would be required to elucidate the complex interactions between Porites associated microbial communities and identify potential pathogenic candidates by fulfilling Koch’s postulates.

We greatly thank Dr. Jean Turquet and Yann Gomard (CRVOI) for their help and guidance in the laboratory. Finally, we would like to address a special thanks to our colleague and great friend Stephanie Bollard (ARVAM). This study would not have been the same without her generous assistance, advice and valuable support.

Additional Information and Declarations

Competing Interests

Author Contributions

Field Study Permissions

Data Availability

The authors declare there are no competing interests.

Mathieu Séré and David A. Wilkinson conceived and designed the experiments, performed the experiments, analyzed the data, contributed reagents/materials/analysis tools, wrote the paper, prepared figures and/or tables, reviewed drafts of the paper.

Michael H. Schleyer, Pascale Chabanet and Jean-Pascal Quod conceived and designed the experiments, wrote the paper, prepared figures and/or tables, reviewed drafts of the paper.

Pablo Tortosa conceived and designed the experiments, performed the experiments, contributed reagents/materials/analysis tools, wrote the paper, prepared figures and/or tables, reviewed drafts of the paper.

The following information was supplied relating to field study approvals (i.e., approving body and any reference numbers):

The sampling of Porites lutea colonies for this study was authorised by the French Department of Ecology, Sustainable Development, Transportation and Housing (DEAL), and CITES (Permit no. FR1197400391-FR1197400394-1).

The following information was supplied regarding data availability:

Cyanobacterial 16S rRNA gene sequences are accessible through the NCBI GeneBank database under accession numbers KF957835–KF957838. Raw 454-pyrosequencing reads were submitted in Study (BioProject PRJNA231011) to the NCBI Sequence Read Archive (SRA).

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
