# Peer review of "Characterisation of an atypical manifestation of black band disease on Porites lutea in the Western Indian Ocean"

_PeerJ, doi:10.7717/peerj.2073_

## Round 0.1 · original submission · Minor Revisions

· Academic Editor

Minor Revisions

Both reviewers were generally positive about your manuscript, and had similar comments on some minor revisions. I have looked over their comments, and find them to be constructive in making your manuscript clearer. Therefore, my decision is 'minor revision".

Reviewer 1 ·

Basic reporting

This article describes coral disease, Black Band Disease (BBD), on Porites corals as PorBBD observed in Reunion Island coral reef, in the Western Indian Ocean. Spatial and seasonal pattern, progression rate, morphological/histological traits, and genetic analysis have been performed. Some results were similar compared to classical/former observations reported, while PorBBD showed different traits especially white front region close to BBD and bacterial composition. Furthermore, such as less content of cyanobacteria, different bacterial communities of PorBBD are of importance. Genetic analysis of bacterial composition and discussion are excellent. This study has important information and will contribute to further study of BBD.

Experimental design

No comments

Validity of the findings

No comments

Additional comments

Séré et al. Characterisation of an atypical manifestation of black band disease on
Porites lutea in the Western Indian Ocean.

Please check the following comments.

Figures and Tables

- Figure 1
Indicate Reunion Island (Fr.) by an arrow or paint Island black, it is too small to find
the Island.
- Figure 3
(ht) (dt) should be (Ht)(Dt), respectively.
- Figure 4
Add explanation for ‘cy’ and ‘cal’, cyanobacterium and calypta (pointed terminal cells)
Scientific names in ‘C’ should be italic.
- Figure 6
Positon of Actinobacteria/Bacterioroides is different in A/B, put correct position.
Scientific names should be italic.
- Table 1
Fill ‘Habitat’ Reef flat or Reef slope.
- Table 2
Line 3: ---healthy tissue => control tissue

References

Some journals are abbreviated other are not.
lines 445, 486: Applied Environmental Microbiology => Appl Environ Microbiol
456: Marine Biology should be Mar Biol
569: Siderastrea siderea => Siderastrea siderea
459,541: abbreviate Journal title
571, 577, 602: abbreviate Journal title

Annotated reviews are not available for download in order to protect the identity of reviewers who chose to remain anonymous.

Reviewer 2 ·

Basic reporting

Adequate descriptions of methods were provided.

Experimental design

1. Line 116: Please clarify how you identified Porites lutea and P. lobata. These species are not distinguishable underwater/in situ; it is necessary to sample skeletal material for examination under the microscope.
2. Line 165: Please describe how you sampled cores from HT, DT and CT.
3. Line 166: Given the high variability in microbial communities between and even within massive Porites surface mucous layers, please justify why you felt that a single core from a single remote control colony was sufficient to provide an adequate description of the microbial community in healthy colonies.

Validity of the findings

1. Lines 303-315: As the microbial consortium within a BBD band is photosynthetic, light limitation is known to be a primary cause of depth limitation in BBD infections. While this has been more rigorously tested in the Caribbean forms, the authors are invited to review work by Sato et al. 2009 and Boyett et al. 2007.

Additional comments

I found the paper well-organized and informative. I was not sure that the description warranted the term “atypical Black Band Disease”, but the authors clearly defined how they were interpreting the disease signs. However, the authors may wish to review a paper by Ainsworth et al. (2007) (Cited in Raymundo and Weil 2016); the term “atypical BBD” has already been used to described something slightly different than what this current paper describes. I have seen this disease myself in several places in the Indo-Pacific and, as many of our coral diseases in this region are poorly described, I believe the paper warrants publication. I am including citations for a couple of references the authors may wish to review that may help round out various points in the discussion and introduction.

Ainsworth T, Kramarsky-Winter E, Loya Y, et al. 2007. Coral disease diagnostics: What’s between a plague and a band? Appl Env Microb 73:981-992.

Raymundo LJ and Weil E. 2016. Chapter 23: Indo-Pacific Colored Band Diseases of Corals. In: CM Woodley, CA Downs, AW Bruckner, JW Porter, SB Galloway (eds). Diseases of Corals. pp. 333-344. Wiley Blackwell.

Boyette H, Bourne D, Willis B. 2007. Elevated temperature and light enhance progression and spread of black band disease on staghorn corals of the Great Barrier Reef. Mar Bio 151: 1711-1720

Sato Y, Bourne D, Willis B. 2009. Dynamics of seasonal outbreaks of black band disease in an assemblage of Montipora species at Pelorus Island (Great Barrier Reef, Australia). Proc Royal Soc Biol Sci 276: 2795-2803.

---

## Round 0.2 · accepted · Accept

· Academic Editor

Accept

The manuscript has been revised well, and is now acceptable for publication.

Please ensure the following small corrections are performed either at the proof stage or earlier:

1. spelling "coralites" changed to "corallites" on lines 133 and 135.
2. delete comma after "...species)" on line 362.
3. line 491: "forth" to "fourth".
4. line 531: "healthy" not "sealthy".
5. line 755: Is this correct? please check "Note the pointed terminal calls called calyptra (cal)".